# Dynamic Rank Factor Model for Text Streams

**Shaobo Han,**[*] **Lin Du,**[*] **Esther Salazar** and **Lawrence Carin**
Duke University, Durham, NC 27708
{shaobo.han, lin.du, esther.salazar, lcarin}@duke.edu

## Abstract

We propose a semi-parametric and dynamic rank factor model for topic modeling, capable of (i) discovering topic prevalence over time, and (ii) learning contemporary multi-scale dependence structures, providing topic and word correlations as a byproduct. The high-dimensional and time-evolving ordinal/rank observations (such as word counts), after an arbitrary monotone transformation, are well accommodated through an underlying dynamic sparse factor model. The framework naturally admits heavy-tailed innovations, capable of inferring abrupt temporal jumps in the importance of topics. Posterior inference is performed through straightforward Gibbs sampling, based on the forward-filtering backward-sampling algorithm. Moreover, an efficient data subsampling scheme is leveraged to speed up inference on massive datasets. The modeling framework is illustrated on two real datasets: the US State of the Union Address and the JSTOR collection from *Science*.

## 1 Introduction

Multivariate longitudinal ordinal/count data arise in many areas, including economics, opinion polls, text mining, and social science research. Due to the lack of discrete multivariate distributions supporting a rich enough correlation structure, one popular choice in modeling correlated categorical data employs the multivariate normal mixture of independent exponential family distributions, after appropriate transformations. Examples include the logistic-normal model for compositional data [1], the Poisson log-normal model for correlated count data [2], and the ordered probit model for multivariate ordinal data [3]. Moreover, a dynamic Bayesian extension of the generalized linear model [4] may be considered, for capturing the temporal dependencies of non-Gaussian data (such as ordinal data). In this general framework, the observations are assumed to follow an exponential family distribution, with natural parameter related to a conditionally Gaussian dynamic model [5], via a nonlinear transformation. However, these model specifications may still be too restrictive in practice, for the following reasons: (i) Observations are usually discrete, non-negative and with a massive number of zero values and, unfortunately, far from any standard parametric distributions (*e.g.*, multinomial, Poisson, negative binomial and even their zero-inflated variants). (ii) The number of contemporaneous series can be large, bringing difficulties in sharing/learning statistical strength and in performing efficient computations. (iii) The linear state evolution is not truly manifested after a nonlinear transformation, where positive shocks (such as outliers and jumps) are magnified and negative shocks are suppressed; hence, handling temporal jumps (up and down) is a challenge for the above models.

We present a flexible semi-parametric Bayesian model, termed *dynamic rank factor model* (DRFM), that does not suffer these drawbacks. We first reduce the effect of model misspecification by modeling the sampling distribution non-parametrically. To do so, we fit the observed data only after some implicit monotone transformation, learned automatically via the extended rank likelihood [6]. Second, instead of treating panels of time series as independent collections of variables, we analyze them jointly, with the high-dimensional cross-sectional dependencies estimated via a latent factor

---

[*]contributed equally

model. Finally, by avoiding nonlinear transformations, both smooth transitions and sudden changes ("jumps") are better preserved in the state-space model, using heavy-tailed innovations.

The proposed model offers an *alternative* to both dynamic and correlated topic models [7, 8, 9], with additional modeling facility of word dependencies, and improved ability to handle jumps. It also provides a semi-parametric Bayesian treatment of dynamic sparse factor model. Further, our proposed framework is applicable in the analysis of multiple ordinal time series, where the innovations follow either stationary Gaussian or heavy-tailed distributions.

## 2 Dynamic Rank Factor Model

We perform analysis of multivariate ordinal time series. In the most general sense, such ordinal variables indicate a ranking of responses in the sample space, rather than a cardinal measure [10]. Examples include real continuous variables, discrete ordered variables with or without numerical scales or, more specially, counts, which can be viewed as discrete variables with integer numeric scales. Our goal is twofold: (i) discover the common trends that govern variations in observations, and (ii) extract interpretable patterns from the cross-sectional dependencies.

Dependencies among multivariate non-normal variables may be induced through normally distributed latent variables. Suppose we have $P$ ordinal-valued time series $y_{p,t}$, $p = 1, \ldots, P$, $t = 1, \ldots, T$. The general framework contains three components:

$$y_{p,t} \sim g(z_{p,t}), \quad z_{p,t} \sim p(\boldsymbol{\theta}_t), \quad \boldsymbol{\theta}_t \sim q(\boldsymbol{\theta}_{t-1}), \tag{1}$$

where $g(\cdot)$ is the sampling distribution, or marginal likelihood for the observations, the latent variable $z_{p,t}$ is modeled by $p(\cdot)$ (assumed to be Gaussian) with underlying system parameters $\boldsymbol{\theta}_t$, and $q(\cdot)$ is the system equation representing Markovian dynamics for the time-evolving parameter $\boldsymbol{\theta}_t$.

In order to gain more model flexibility and robustness against misspecification, we propose a semi-parametric Bayesian dynamic factor model for multiple ordinal time series analysis. The model is based on the *extended rank likelihood* [6], allowing the transformation from the latent conditionally Gaussian dynamic model to the multivariate observations, treated non-parametrically.

**Extended rank likelihood (ERL):** There exist many approaches for dealing with ordinal data, however, they all have some restrictions. For continuous variables, the underlying normality assumption could be easily violated without a carefully chosen deterministic transformation. For discrete ordinal variables, an ordered probit model, with cut points, becomes computationally expensive if the number of categories is large. For count variables, a multinomial model requires finite support on the integer values. Poisson and negative binomial models lack flexibility from a practical viewpoint, and often lead to non-conjugacy when employing log-normal priors.

Being aware of these issues, a natural candidate for consideration is the ERL [6]. With appropriate monotone transformations learned automatically from data, it offers a unified framework for handling both continuous [11] and discrete ordinal variables. The ERL depends only on the ranks of the observations (zero values in observations are further restricted to have negative latent variables),

$$z_{p,t} \in D(\boldsymbol{Y}) \equiv \{\boldsymbol{z}_{p,t} \in \mathbb{R} : y_{p,t} < y_{p',t'} \Rightarrow z_{p,t} < z_{p',t'}, \text{and } z_{p,t} \leq 0 \text{ if } y_{p,t} = 0\}. \tag{2}$$

In particular, this offers a distribution-free approach, with relaxed assumptions compared to parametric models, such as Poisson log-normal [12]. It also avoids the burden of computing nuisance parameters in the ordered probit model (cut points). The ERL has been utilized in Bayesian Gaussian copula modeling, to characterize the dependence of mixed data [6]. In [13] a low-rank decomposition of the covariance matrix is further employed and efficient posterior sampling is developed in [14]. The proposed work herein can be viewed as a dynamic extension of that framework.

### 2.1 Latent sparse dynamic factor model

In the forthcoming text, $\mathcal{G}(\alpha, \beta)$ denotes a gamma distribution with shape parameter $\alpha$ and rate parameter $\beta$, $\text{TN}_{(l,u)}(\mu, \sigma^2)$ denotes a univariate truncated normal distribution within the interval $(l, u)$, and $\mathcal{N}_+(0, \sigma^2)$ is the half-normal distribution that only has non-negative support.

Assume $\boldsymbol{z}_t \sim \mathcal{N}(\boldsymbol{0}, \boldsymbol{\Omega}_t)$, where $\boldsymbol{\Omega}_t$ is usually a high-dimensional ($P \times P$) covariance matrix. To reduce the number of parameters, we assume a low rank factor model decomposition of the covariance matrix $\boldsymbol{\Omega}_t = \boldsymbol{\Lambda} \boldsymbol{V}_t \boldsymbol{\Lambda}^T + \boldsymbol{R}$ such that

$$\boldsymbol{z}_t = \boldsymbol{\Lambda} \boldsymbol{s}_t + \boldsymbol{\epsilon}_t, \quad \boldsymbol{\epsilon}_t \sim \mathcal{N}(\boldsymbol{0}, \boldsymbol{R}), \quad \boldsymbol{R} = \boldsymbol{I}_P. \tag{3}$$

Common trends (importance of topics) are captured by a low-dimensional factor score parameter $s_t$. We assume autoregressive dynamics on $s_{k,t} \leftarrow \mathrm{AR}(1|(\rho_k, \delta_{k,t}))$ with heavy-tailed innovations,

$$s_{k,t} = \rho_k s_{k,t-1} + \delta_{k,t}, \quad 0 < \rho_k < 1, \quad \delta_{k,t} \sim \mathrm{TPBN}(e, f, \nu), \quad \nu^{1/2} \sim \mathcal{C}^+(0, h), \quad (4)$$

where $\delta_{k,t}$ follows the *three-parameter beta mixture of normal* $\mathrm{TPBN}(e, f, \nu)$ distribution [15]. Parameter $e$ controls the peak around zero, $f$ controls the heaviness on the tails, and $\nu$ controls the global sparsity with a half-Cauchy prior [16]. This prior encourages smooth transitions in general, while jumps are captured by the heavy tails. The conjugate hierarchy may be equivalently represented as

$$\delta_{k,t} \sim \mathcal{N}(0, \tau_{k,t}), \quad \tau_{k,t} \sim \mathcal{G}(e, \eta_{k,t}), \quad \eta_{k,t} \sim \mathcal{G}(f, \nu) \quad \nu \sim \mathcal{G}(1/2, \zeta), \quad \zeta \sim \mathcal{G}(1/2, h^2).$$

Truncated normal priors are employed on $\rho_k$, $\rho_k \sim \mathrm{TN}_{(0,1)}(\mu_0, \sigma_0^2)$, and assume $s_{0,k} \sim \mathcal{N}(0, \sigma_s^2)$. Note that the extended rank likelihood is scale-free; therefore, we do not need to include a redundant intercept parameter in (3). For the same reason, we set $\boldsymbol{R} = \boldsymbol{I}_P$.

**Model identifiability issues:** Although the covariance matrix $\boldsymbol{\Omega}_t$ is not identifiable [10], the related correlation matrix $\boldsymbol{C}_t = \Omega_{[i,j],t}/\sqrt{\Omega_{[i,i],t}\Omega_{[j,j],t}}$, $(i, j = 1, \ldots, P)$ may be identified, using the parameter expansion technique [3, 13]. Further, the rank $K$ in the low-rank decomposition of $\boldsymbol{\Omega}_t$ is also not unique. For the purpose of brevity, we do not explore this uncertainty here, but the tools developed in the Bayesian factor analysis literature [17, 18, 19] can be easily adopted.

Identifiability is a key concern for factor analysis. Conventionally, for fixed $K$, a full-rank, lower-triangular structure in $\boldsymbol{\Lambda}$ ensures identifiability [20]. Unfortunately, this assumption depends on the ordering of variables. As a solution, we add nonnegative and sparseness constraints on the factor loadings, to alleviate the inherit ambiguity, while also improving interpretability. Also, we add a Procrustes post-processing step [21] on the posterior samples, to reduce this indeterminacy.

The nonnegative and (near) sparseness constraints are imposed by the following hierarchy,

$$\lambda_{p,k} \sim \mathcal{N}_+(0, l_{p,k}) \quad l_{p,k} \sim \mathcal{G}(a, u_{p,k}), \quad u_{p,k} \sim \mathcal{G}(b, \phi_k), \quad \phi_k^{1/2} \sim \mathcal{C}^+(0, d). \quad (5)$$

Integrating out $l_{p,k}$ and $u_{p,k}$, we obtain a half-TPBN prior $\lambda_{p,k} \sim \mathrm{TPBN}_+(a, b, \phi_k)$. The column-wise shrinkage parameters $\phi_k$ enable factors to be of different sparsity levels [22]. We set hyperparameters $a = b = e = f = 0.5$, $d = P$, $h = 1$, $\sigma_s^2 = 1$. For weakly informative priors, we set $\alpha = \beta = 0.01$; $\mu_0 = 0.5$, $\sigma_0^2 = 10$.

## 2.2 Extension to handle multiple documents

At each time point $t$ we may have a corpus of documents $\{\boldsymbol{y}_t^{n_t}\}_{n_t=1}^{N_t}$, where $\boldsymbol{y}_t^{n_t}$ is a $P$-dimensional observation vector, and $N_t$ denotes the number of documents at time $t$. The model presented in Section 2.1 is readily extended to handle this situation. Specifically, at each time point $t$, for each document $n_t$, the ERL representation for word count $p$, denoted by $y_{p,t}^{n_t}$, is

$$y_{p,t}^{n_t} = g\left(z_{p,t}^{n_t}\right), \quad p = 1, \ldots, P, \quad t = 1, \ldots, T, \quad n_t = 1, \ldots, N_t,$$

where $\boldsymbol{z}_t^{n_t} \in \mathbb{R}^P$ and $P$ is the vocabulary size. We assume a latent factor model for $\boldsymbol{z}_t^{n_t}$ such that

$$\boldsymbol{z}_t^{n_t} = \boldsymbol{\Lambda} \boldsymbol{b}_t^{n_t} + \boldsymbol{\epsilon}_t^{n_t}, \quad \boldsymbol{\epsilon}_t^{n_t} \sim \mathcal{N}(\boldsymbol{0}, \boldsymbol{I}_P), \quad \boldsymbol{b}_t^{n_t} \sim \mathcal{N}(\boldsymbol{s}_t, \boldsymbol{\Gamma}), \quad \boldsymbol{\Gamma} = \mathrm{diag}(\boldsymbol{\gamma}), \quad \gamma_k^{-1} \sim \mathcal{G}(\alpha, \beta),$$

where $\boldsymbol{\Lambda} \in \mathbb{R}_+^{P \times K}$ is the topic-word loading matrix, representing the $K$ topics as columns of $\boldsymbol{\Lambda}$. The factor score vector $\boldsymbol{b}_t^{n_t} \in \mathbb{R}^K$ is the topic usage for each document $\boldsymbol{y}_t^{n_t}$, corresponding to locations in a low-dimensional $\mathbb{R}^K$ space. The other parts of the model remain unchanged. The latent trajectory $\boldsymbol{s}_{1:T}$ represents the common trends for the $K$ topics. Moreover, through the forward filtering backward sampling (FFBS) algorithm [23, 24], we also obtain time-evolving topic correlation matrices $\boldsymbol{\Phi}_t \in \mathbb{R}^{K \times K}$ and word dependencies matrices $\boldsymbol{C}_t \in \mathbb{R}^{P \times P}$, offering a multi-scale graph representation, a useful tool for document visualization.

## 2.3 Comparison with admixture topic models

Many topic models are unified in the admixture framework [25],

$$\underset{\mathrm{Admix}}{\mathrm{P}}(\boldsymbol{y}_n|\boldsymbol{w}, \boldsymbol{\Phi}) = \underset{\mathrm{Base}}{\mathrm{P}}\left(\boldsymbol{y}_n \,\middle|\, \overline{\boldsymbol{\phi}}_n = \sum_{k=1}^{K} w_{k,n} \boldsymbol{\phi}_k\right), \quad (6)$$

where $\boldsymbol{y}_n$ is the $P$-dimensional observation vector of word counts in the $n$ th document, and $P$ denotes the vocabulary size. Traditionally, $\boldsymbol{y}_n$ is generated from an admixture of base distributions, $\boldsymbol{w}_n$ is the admixture weight (topic proportion for document $n$), and $\boldsymbol{\phi}_k$ is the canonical parameter (word

distribution for topic $k$), which denotes the location of the $k$th topic on the $P$-1 dimensional simplex. For example, latent Dirichlet allocation (LDA) [26] assumes the base distribution to be multinomial, with $\phi_k \sim \mathrm{Dir}(\boldsymbol{\alpha}_0)$, $\boldsymbol{w}_n \sim \mathrm{Dir}(\boldsymbol{\beta}_0)$. The correlated topic model (CTM) [8] modifies the topic distribution, with $\boldsymbol{w}_n \sim \mathrm{Logistic\ Normal}(\boldsymbol{\mu}, \boldsymbol{\Sigma})$. The dynamic topic model (DTM) [7] analyzes document collections in a known chronological order. In order to incorporate the state space model, both the topic proportion and the word distribution are changed to logistic normal, with isotropic covariance matrices $\boldsymbol{w}_t \sim \mathrm{Logistic\ Normal}(\boldsymbol{w}_{t-1}, \sigma^2 \boldsymbol{I}_K)$ and $\boldsymbol{\phi}_{k,t} \sim \mathrm{Logistic\ Normal}(\boldsymbol{\phi}_{k,t-1}, v\boldsymbol{I}_P)$, respectively. To overcome the drawbacks of multinomial base, spherical topic models [27] assume the von Mises-Fisher (vMF) distribution as its base distribution, with $\phi_k \sim \mathrm{vMF}(\boldsymbol{\mu}, \xi)$ lying on a unit $P$-1 dimensional sphere. Recently in [25] the base and word distribution are both replaced with Poisson Markov random fields (MRFs), which characterizes word dependencies.

We present here a semi-parametric factor model formulation,

$$\mathrm{P}(\boldsymbol{y}_n | \boldsymbol{s}, \boldsymbol{\Lambda}) \triangleq \mathrm{P}\left( \boldsymbol{z}_n \in D(\boldsymbol{Y}) \,\middle|\, \overline{\boldsymbol{\lambda}}_n = \sum_{k=1}^K s_{k,n} \boldsymbol{\lambda}_k \right), \tag{7}$$

with $\boldsymbol{y}_n$ defined as above, $\boldsymbol{\lambda}_k \in \mathbb{R}_+^P$ is a vector of nonnegative weights, indicating the $P$ vocabulary usage in each individual topics $k$, and $\boldsymbol{s}_n \in \mathbb{R}^K$ is the topic usage. Note that the extended rank likelihood does not depend on any assumptions about the data marginal distribution, making it appropriate for a broad class of ordinal-valued observations, $e.g.$, term frequency-inverse document frequency (tf-idf) or rankings, beyond word counts. However, the proposed model here is not an admixture model, as the topic usage is allowed to be either positive or negative.

The DRFM framework has some appealing advantages: (i) It is more natural and convenient to incorporate with sparsity, rank selection, and state-space model; (ii) it provides topic-correlations and word-dependences as a byproduct; and (iii) computationally, this model is tractable and often leads to locally conjugate posterior inference. DRFM has limitations. Since the marginal distributions are of unspecified types, objective criteria (e.g. perplexity) is not directly computable. This makes quantitative comparisons to other parametric baselines developed in the literature very difficult.

## 3 Conjugate Posterior Inference

Let $\boldsymbol{\Theta} = \{\boldsymbol{\Lambda}, \boldsymbol{S}, \boldsymbol{L}, \boldsymbol{U}, \phi, \boldsymbol{\omega}, \boldsymbol{\rho}, \boldsymbol{\tau}, \boldsymbol{\eta}, \nu, \zeta\}$ denote the set of parameters in basic model, and let $\boldsymbol{Z}$ be the augmented data (from the ERL). We use Gibbs sampling to approximate the joint posterior distribution $p(\boldsymbol{Z}, \boldsymbol{\Theta} | \boldsymbol{Z} \in R(\boldsymbol{Y}))$. The algorithm alternates between sampling $p(\boldsymbol{Z} | \boldsymbol{\Theta}, \boldsymbol{Z} \in R(\boldsymbol{Y}))$ and $p(\boldsymbol{\Theta} | \boldsymbol{Z}, \boldsymbol{Z} \in R(\boldsymbol{Y}))$ (reduced to $p(\boldsymbol{\Theta} | \boldsymbol{Z})$). The derivation of the Gibbs sampler is straightforward, and for brevity here we only highlight the sampling steps for $\boldsymbol{Z}$, and the forward filtering backward sampling (FFBS) steps for the trajectory $\boldsymbol{s}_{1:T}$. The Supplementary Material contains further details for the inference.

- Sampling $z_{p,t}$: $p(z_{p,t} | \boldsymbol{\Theta}, \boldsymbol{Z} \in R(\boldsymbol{Y}), \boldsymbol{Z}_{-p,-t}) \sim \mathrm{TN}_{[\underline{z_{p,t}}, \overline{z_{p,t}}]}(\sum_{k=1}^K \lambda_{p,k} s_{k,t}, 1)$, where $\underline{z_{p,t}} = \max\{z_{p',t'} : y_{p',t'} < y_{p,t}\}$ and $\overline{z_{p,t}} = \min\{z_{p',t'} : y_{p',t'} > y_{p,t}\}$.

This conditional sampling scheme is widely used in [6, 10, 13]. In [14] a novel Hamiltonian Monte Carlo (HMC) approach has been developed recently, for a Gaussian copula extended rank likelihood model, where ranking is only within each row of $\boldsymbol{Z}$. This method simultaneously samples a column vector of $\boldsymbol{z}_i$ conditioned on other columns $\boldsymbol{Z}_{-i}$, with higher computation but better mixing.

- Sampling $\boldsymbol{s}_t$: we have the state model $\boldsymbol{s}_t | \boldsymbol{s}_{t-1} \sim \mathcal{N}(\boldsymbol{A}\boldsymbol{s}_{t-1}, \boldsymbol{Q}_t)$, and the observation model $\boldsymbol{z}_t | \boldsymbol{s}_t \sim \mathcal{N}(\boldsymbol{\Lambda}\boldsymbol{s}_t, \boldsymbol{R})$,[1] where $\boldsymbol{A} = \mathrm{diag}(\boldsymbol{\rho})$, $\boldsymbol{Q}_t = \mathrm{diag}(\boldsymbol{\tau}_t)$, $\boldsymbol{R} = \boldsymbol{I}_P$. for $t = 1, \dots, T$
  1. Forward Filtering: beginning at $t = 0$ with $\boldsymbol{s}_0 \sim \mathcal{N}(\boldsymbol{0}, \sigma_s^2 \boldsymbol{I}_K)$, for all $t = 1, \dots, T$, we find the on-line posteriors at $t$, $p(\boldsymbol{s}_t | \boldsymbol{z}_{1:t}) = \mathcal{N}(\boldsymbol{m}_t, \boldsymbol{V}_t)$, where $\boldsymbol{m}_t = \boldsymbol{V}_t \{\boldsymbol{\Lambda}^T \boldsymbol{R}^{-1} \boldsymbol{z}_t + \boldsymbol{H}_t^{-1} \boldsymbol{A} \boldsymbol{m}_{t-1}\}$, $\boldsymbol{V}_t = [\boldsymbol{H}_t^{-1} + \boldsymbol{\Lambda}^T \boldsymbol{R}^{-1} \boldsymbol{\Lambda}]^{-1}$, and $\boldsymbol{H}_t = \boldsymbol{Q}_t + \boldsymbol{A}\boldsymbol{V}_{t-1}\boldsymbol{A}^T$.
  2. Backward Sampling: starting from $\mathcal{N}(\widetilde{\boldsymbol{m}}_t, \widetilde{\boldsymbol{V}}_t)$, the backward smoothing density, $i.e.$, the conditional distribution of $\boldsymbol{s}_{t-1}$ given $\boldsymbol{s}_t$, is $p(\boldsymbol{s}_{t-1} | \boldsymbol{s}_t, \boldsymbol{z}_{1:(t-1)}) = \mathcal{N}(\widetilde{\boldsymbol{\mu}}_{t-1}, \widetilde{\boldsymbol{\Sigma}}_{t-1})$, where $\widetilde{\boldsymbol{\mu}}_{t-1} = \widetilde{\boldsymbol{\Sigma}}_{t-1}\{\boldsymbol{A}^T \boldsymbol{Q}_t^{-1} \boldsymbol{s}_t + \boldsymbol{V}_{t-1}^{-1} \boldsymbol{m}_{t-1}\}$, $\widetilde{\boldsymbol{\Sigma}}_{t-1} = (\boldsymbol{V}_{t-1}^{-1} + \boldsymbol{A}^T \boldsymbol{Q}_t^{-1} \boldsymbol{A})^{-1}$.

There exist different variants of FFBS schemes (see [28] for a detailed comparison); the method we choose here enjoys fast decay in autocorrelation and reduced computation time.

### 3.1 Time-evolving topic and word dependencies

We also have the backward recursion density at $t-1$, $p(s_{t-1}|z_{1:T}) = \mathcal{N}(\widetilde{m}_{t-1}, \widetilde{V}_{t-1})$, where $\widetilde{m}_{t-1} = \widetilde{\Sigma}_{t-1}(A^T Q_t^{-1}\widetilde{m}_t + V_{t-1}^{-1}m_{t-1})$ and $\widetilde{V}_{t-1} = \widetilde{\Sigma}_{t-1} + \widetilde{\Sigma}_{t-1}A^T Q_t^{-1}\widetilde{V}_t Q_t^{-1}A\widetilde{\Sigma}_{t-1}$. We perform inference on the $K \times K$ time-evolving topic dependences in $s_{1:T}$, using the posterior covariances $\{\widehat{V}_{1:T}\}$ (with topic correlation matrices $\Phi_{1:T}$, $\Phi_{[r,s],t} = V_{[r,s],t}/\sqrt{V_{[r,r],t}V_{[s,s],t}}$, $r,s = 1,\ldots,K$), and further obtain the $P \times P$ time-evolving word dependencies capsuled in $\{\Omega_{1:T}\}$ with $\Omega_t = \Lambda\widetilde{V}_t\Lambda^T + I_P$. Essentially, this can be viewed as a dynamic Gaussian copula model, $y_{p,t} = g(\widetilde{z}_{p,t})$, $\widetilde{z}_t \sim \mathcal{N}(0, C_t)$, where $g(\cdot)$ is a non-decreasing function of a univariate marginal likelihood and $C_t$ $(t = 1,\ldots,T)$ is the correlation matrix capturing the multivariate dependence. We obtain a posterior distribution for $C_{1:T}$ as a byproduct, without having to estimate the nuisance parameters in marginal likelihoods $g(\cdot)$. This decoupling strategy resembles the idea of copula models.

### 3.2 Accelerated MCMC via document subsampling

For large-scale datasets, recent approaches efficiently reduce the computational load of Monte Carlo Markov chain (MCMC) by data subsampling [29, 30]. We borrow this idea of subsampling documents when considering a large corpora (*e.g.*, in our experiments, we consider analysis of articles in the magazine *Science*, composed of 139379 articles from years 1880 to 2002, and a vocabulary size 5855). In our model, the augmented data $z_t^{n_t}$ $(n_t = 1,\ldots,N_t)$ for each document is relatively expensive to sample. One simple method is random document sampling without replacement. However, by treating all likelihood contributions symmetrically, this method leads to a highly inefficient MCMC chain with poor mixing [29].

Alternatively, we adopt the probability proportional-to-size (PSS) sampling scheme in [30], *i.e.*, sampling the documents with inclusion probability proportional to the likelihood contributions. For each MCMC iteration, the sub-sampling procedure for documents at time $t$ is designed as follows:

- **Step 1:** Given a small subset $\mathcal{V}_t \subset \{1,\ldots,N_t\}$ of chosen documents, only sample $\{z_t^d\}$ for all $d \in \mathcal{V}_t$ and compute the augment log-likelihood contributions (with $B_t$ integrated out) $\ell_{\mathcal{V}_t}(z_t^d) = \mathcal{N}(\Lambda s_t, \widetilde{R})$, where $\widetilde{R} = \Lambda\Gamma\Lambda^T + I_P$. Note that, only a $K$-dimensional matrix inversion is required, by using the Woodbury matrix inversion formula $\widetilde{R}^{-1} = I_P - \Lambda(\Gamma^{-1} + \Lambda^T\Lambda)^T\Lambda^T$.

- **Step 2:** Similar to [30], we use a Gaussian process [31] to predict the log-likelihood for the remaining documents $\ell_{\mathcal{V}_t^c}(z_t^d) = \mathcal{K}(\mathcal{V}_t^c, \mathcal{V}_t)\mathcal{K}(\mathcal{V}_t, \mathcal{V}_t)^{-1}\ell_{\mathcal{V}_t}(z_t^d)$, where $\mathcal{K}$ is a $N_t \times N_t$ squared-exponential kernel, which denotes the similarity of documents: $\mathcal{K}(y_t^i, y_t^j) = \sigma_f^2\exp\left(-||y_t^i - y_t^j||^2/(2s^2)\right)$, $i,j = 1,\ldots,N_t$, $\sigma_f^2 = 1$, $s = 1$.

- **Step 3:** Calculate the inclusion probability $w_d \propto \exp[\ell(z_t^d)]$, $d = 1,\ldots,N_t$, $\widetilde{w}_d = w_d/\sum_{d'} w_{d'}$.

- **Step 4:** Sampling the next subset $\mathcal{V}_t$ of pre-specified size $|\mathcal{V}_t|$ with inclusion probability $\widetilde{w}_d$, and store it for the use of the next MCMC iteration.

In practice, this adaptive design allows MCMC to run more efficiently on a full dataset of large scale, often mitigating the need to do parallel MCMC implementation. Future work could also consider nonparametric function estimation subject to monotonicity constraint, e.g. Gaussian process projections recently developed in [32].

## 4 Experiments

Different from DTM [7] , the proposed model has the jumps directly at the level of the factor scores (no exponentiation or normalization needed), and therefore it proved more effective in uncovering jumps in factor scores over time. Demonstrations of this phenomenon in a synthetic experiment are detailed in the Supplementary Material. In the following, we present exploratory data analysis on two real examples, demonstrating the ability of the proposed model to infer temporal jumps in topic importance, and to infer correlations across topics and words.

### 4.1 Case Study I: State of the Union dataset

The State of the Union dataset contains the transcripts of $T = 225$ US State of the Union addresses, from 1790 to 2014. We take each transcript as a document, *i.e.*, we have one document per year.

After removing stop words, and removing terms that occur fewer than 3 times in one document and less than 10 times overall, we have $P = 7518$ unique words. The observation $y_{p,t}$ corresponds to the frequency of word $p$ of the State of the Union transcript from year $t$.

We apply the proposed DRFM setting and learned $K = 25$ topics. To better understand the temporal dynamic per topic, six topics are selected and the posterior mean of their latent trajectories $s_{k,1:T}$ are shown in Figure 1 (with also the top 12 most probable words associated with each of the topics). A complete table with all 25 learned topics and top 12 words is provided in the Supplementary Material. The learned trajectory associated with every topic indicates different temporal patterns across all the topics. Clearly, we can identify jumps associated with some key historical events. For instance, for Topic 10, we observe a positive jump in 1846 associated with the Mexican-American war. Topic 13 is related with the Spanish-American war of 1898, with a positive jump in that year. In Topic 24, we observe a positive jump in 1914, when the Panama Canal was officially opened (words *Panana* and *canal* are included). In Topic 18, the positive jumps observed from 1997 to 1999 seem to be associated with the creation of the State Children's Health Insurance Program in 1997. We note that the words for this topic are explicitly related with this issue. Topic 25 appears to be related to banking; the significant spike around 1836 appears to correspond to the Second Bank of the United States, which was allowed to go out of existence, and end national banking that year. In 1863 Congress passed the National Banking Act, which ended the "free-banking" period from 1836-1863; note the spike around 1863 in Topic 25.

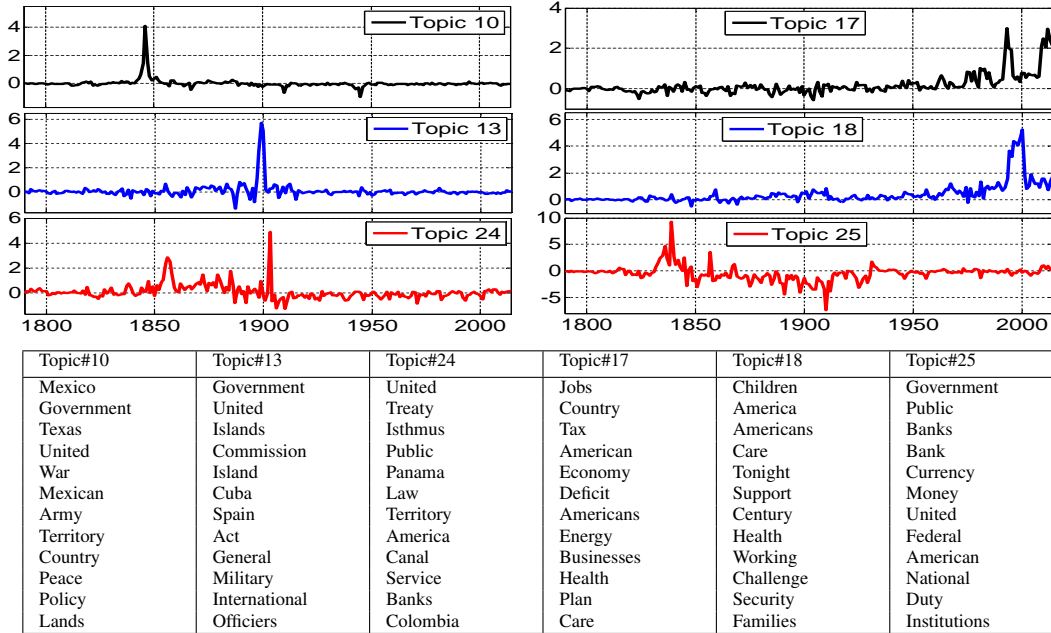

| Topic#10 | Topic#13 | Topic#24 | Topic#17 | Topic#18 | Topic#25 |
|---|---|---|---|---|---|
| Mexico | Government | United | Jobs | Children | Government |
| Government | United | Treaty | Country | America | Public |
| Texas | Islands | Isthmus | Tax | Americans | Banks |
| United | Commission | Public | American | Care | Bank |
| War | Island | Panama | Economy | Tonight | Currency |
| Mexican | Cuba | Law | Deficit | Support | Money |
| Army | Spain | Territory | Americans | Century | United |
| Territory | Act | America | Energy | Health | Federal |
| Country | General | Canal | Businesses | Working | American |
| Peace | Military | Service | Health | Challenge | National |
| Policy | International | Banks | Plan | Security | Duty |
| Lands | Officiers | Colombia | Care | Families | Institutions |

Figure 1: (*State of the Union* **dataset**) Above: Time evolving from 1790 to 2014 for six selected topics. The plotted values represent the posterior means. Below: Top 12 most probable words associated with the above topics.

Our modeling framework is able to capture dynamic patterns of topics and word correlations. To illustrate this, we select three years (associated with some meaningful historical events) and analyze their corresponding topic and word correlations. Figure 2 (first row) shows graphs of the topic correlation matrices, in which the nodes represent topics and the edges indicate positive (green) and negative (red) correlations (we show correlations with absolute value larger than 0.01). We notice that Topics 11 and 22 are positively correlated with those years. Some of the most probable words associated with each of them are: *increase, united, law and legislation* (for Topic 11) and *war, Mexico, peace, army, enemy and military* (for Topic 22). We also are interested in understanding the time-varying correlation between words. To do so, and for the same years as before, in Figure 2 (second row) we plot the dendrogram associated with the learned correlation matrix for words. In the plots, different colors indicate highly correlated word clusters defined by cutting the branches off the dendrogram. Those figures reveal different sets of highly correlated words for different years. By

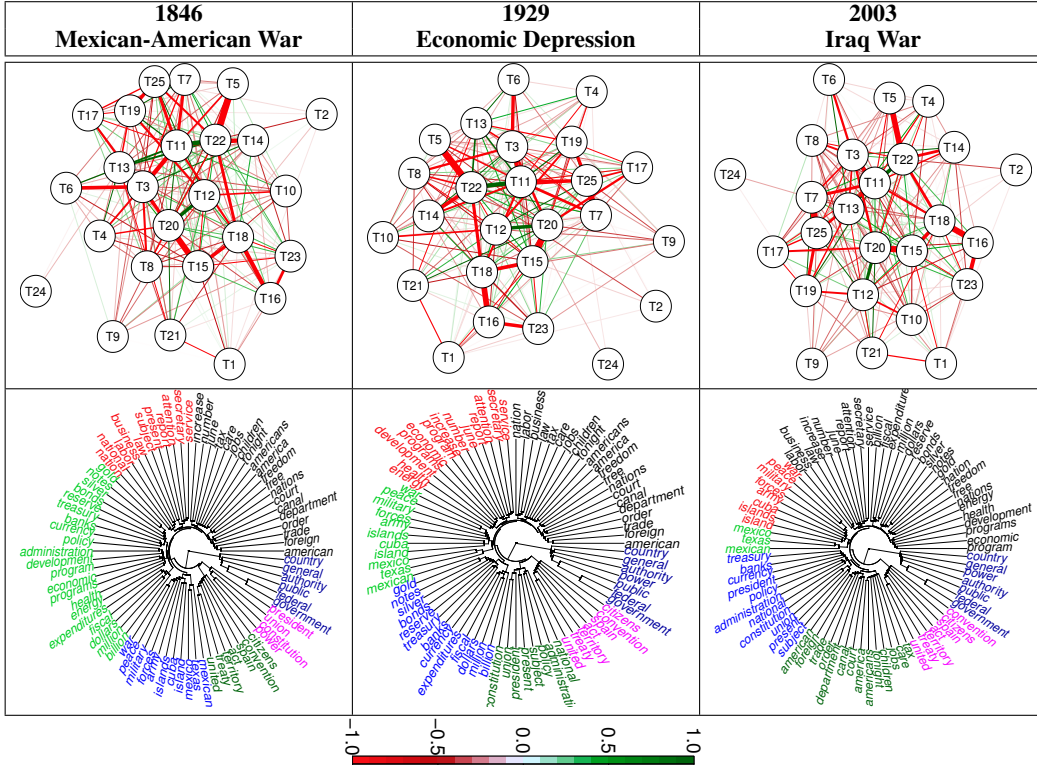

Figure 2: (**State of the Union** dataset) First row: Inferred correlations between topics for some specific years associated with some meaningful historical events. Green edges indicate positive correlations and red edges indicate negative correlations. Second row: Learned dendrogram based upon the correlation matrix between the top 10 words associated with each topic (we display 80 unique words in total).

inspecting all the words correlation, we noticed that the set of words {*government, federal, public, power, authority, general, country*} are highly correlated across the whole period.

## 4.2 Case Study II: Analysis of *Science* dataset

We analyze a collection of scientific documents from the JSTOR *Science* journal [7]. This dataset contains a collection of 139379 documents from 1880 to 2002 ($T = 123$), with approximately 1100 documents per year. After removing terms that occurred fewer than 25 times, the total vocabulary size is $P = 5855$. We learn $K = 50$ topics from the inferred posterior distribution, for brevity and simplicity, we only show 20 of them. We handle about 2700 documents per iteration (subsampling rate: 2%). Table 1 shows the 20 selected topics and the top 10 most probable words associated with each of them. By inspection, we notice that those topics are related with specific fields in science. For instance, Topic 2 is more related to "scientific research", Topic 10 to "natural resources", and Topic 15 to "genetics". Figure 3 shows the time-varying trend for some specific words, $\widehat{z}_{p,1:T}$, which reveals the *importance* of those words across time. Finally, Figure 4 shows the correlation between the selected 20 topics. For instance, in 1950 and 2000, topic 9 (related to *mouse, cells, human, transgenic*) and topic 17 (related to *virus, rna, tumor, infection*) are highly correlated.

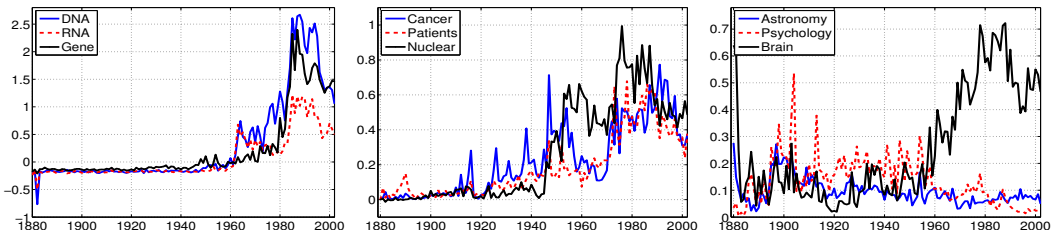

Figure 3: (**Science** dataset) the inferred latent trend for variable $\widehat{z}_{p,1:T}$ associated with words.

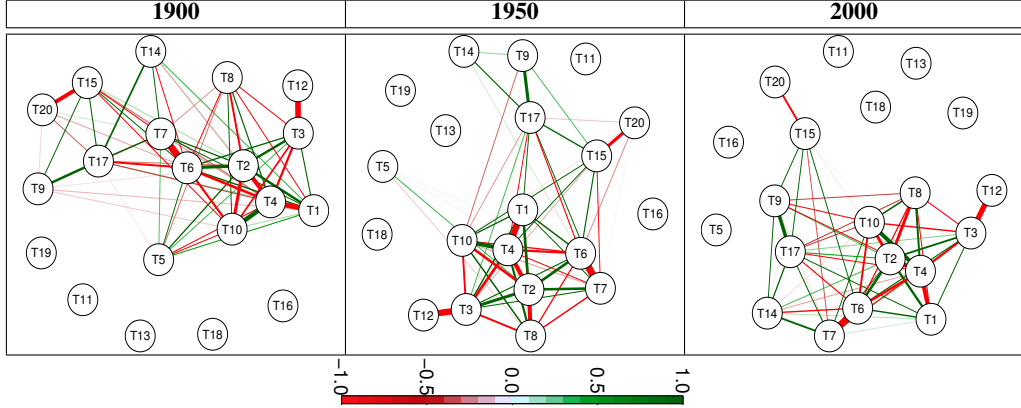

Figure 4: (*Science* **dataset**) Inferred correlations between topics for some specific years. Green edges indicate positive correlations and red edges indicate negative correlations.

Table 1: Selected 20 topics associated with the analysis of the *Science* dataset and top 10 most probable words.

| Topic#1 | Topic#2 | Topic#3 | Topic#4 | Topic#5 | Topic#6 | Topic#7 | Topic#8 | Topic#9 | Topic#10 |
|---|---|---|---|---|---|---|---|---|---|
| cells | research | field | animals | energy | university | science | work | mice | water |
| cell | national | magnetic | brain | oil | professor | scientific | research | mouse | surface |
| normal | government | solar | neurons | percent | college | new | scientific | type | temperature |
| two | support | energy | activity | production | president | scientists | laboratory | wild | soil |
| growth | federal | spin | response | fuel | department | human | made | fig | pressure |
| development | development | state | rats | total | research | men | university | cells | sea |
| tissue | new | electron | control | growth | institute | sciences | results | human | plants |
| body | program | quantum | fig | states | director | knowledge | science | transgenic | solution |
| egg | scientific | temperature | effects | electricity | society | meeting | survey | animals | plant |
| blood | basic | current | days | coal | school | work | department | mutant | air |

| Topic#11 | Topic#12 | Topic#13 | Topic#14 | Topic#15 | Topic#16 | Topic#17 | Topic#18 | Topic#19 | Topic#20 |
|---|---|---|---|---|---|---|---|---|---|
| system | energy | association | protein | human | professor | virus | energy | stars | rna |
| nuclear | theory | science | proteins | genome | university | rna | electron | mass | fig |
| new | temperature | meeting | cell | sequence | society | viruses | state | star | mrna |
| systems | radiation | university | membrane | chromosome | department | particles | fig | temperature | protein |
| power | atoms | american | amino | gene | college | tumor | two | solar | site |
| cost | surface | society | sequence | genes | president | mice | structure | gas | sequence |
| computer | atomic | section | binding | map | director | disease | reaction | data | splicing |
| fuel | mass | president | acid | data | american | viral | laser | density | synthesis |
| coal | atom | committee | residues | sequences | appointed | human | high | surface | trna |
| plant | time | secretary | sequences | genetic | medical | infection | temperature | galaxies | rnas |

## 5   Discussion

We have proposed a DRFM framework that could be applied to a broad class of applications such as: (i) dynamic topic model for the analysis of time-stamped document collections; (ii) joint analysis of multiple time series, with ordinal valued observations; and (iii) multivariate ordinal dynamic factor analysis or dynamic copula analysis for mixed type of data. The proposed model is a semi-parametric methodology, which offers modeling flexibilities and reduces the effect of model mis-specification. However, as the marginal likelihood is distribution-free, we could not calculate the model evidence or other evaluation metrics based on it (e.g. held-out likelihood). As a consequence, we are lack of objective evaluation criteria, which allow us to perform formal model comparisons. In our proposed setting, we are able to perform either retrospective analysis or multi-step ahead forecasting (using the recursive equations derived in the FFBS algorithm). Finally, our inference framework is easily adaptable for using sequential Monte Carlo (SMC) methods [33] allowing on-line learning.

**Acknowledgments**

The research reported here was funded in part by ARO, DARPA, DOE, NGA and ONR. The authors are grateful to Jonas Wallin, Lund University, Sweden, for providing efficient package on simulation of the GIG distribution.

## Footnotes

[1]For brevity, we omit the dependencies on $\boldsymbol{\Theta}$ in notation

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
