[Supplementary Material]

# Dynamic Rank Factor Model for Text Streams Supplementary Material

**Shaobo Han,**[*] **Lin Du,**[*] **Esther Salazar and Lawrence Carin**
Duke University, Durham, NC 27708
{shaobo.han, lin.du, esther.salazar, lcarin}@duke.edu

## 1 Gibbs Sampling for the Basic Model

### 1.1 Model

$$y_{p,t} = g\left(z_{p,t}\right) \tag{1}$$

$$\boldsymbol{z}_t = \boldsymbol{\Lambda}\boldsymbol{s}_t + \boldsymbol{\epsilon}_t, \quad \boldsymbol{\epsilon}_t \sim \mathcal{N}(\boldsymbol{0}, \boldsymbol{R}), \quad \boldsymbol{R} = \boldsymbol{I}_P \tag{2}$$

$$\lambda_{p,k} \sim \text{TPBN}_+(a, b, \phi_k), \quad \phi_k^{1/2} \sim \mathcal{C}^+(0, d) \tag{3}$$

$$s_{k,t} = \rho_k s_{k,t-1} + \delta_{k,t}, \quad 0 < \rho_k < 1 \tag{4}$$

$$\delta_{k,t} \sim \text{TPBN}(e, f, \nu), \quad \nu^{1/2} \sim \mathcal{C}^+(0, h) \tag{5}$$

$$\rho_k \sim \text{TN}_{(0,1)}(\mu_0, \sigma_0^2), \quad s_{0,k} \sim \mathcal{N}(0, \sigma_s^2) \tag{6}$$

### 1.2 Conjugate hierarchy

$$\lambda_{p,k} \sim \mathcal{N}_+(0, l_{p,k}) \quad l_{p,k} \sim \mathcal{G}(a, u_{p,k}), \quad u_{p,k} \sim \mathcal{G}(b, \phi_k) \tag{7}$$

$$\phi_k \sim \mathcal{G}(1/2, \omega_k), \quad \omega_k \sim \mathcal{G}(1/2, d^2) \tag{8}$$

$$\delta_{k,t} \sim \mathcal{N}(0, \tau_{k,t}), \quad \tau_{k,t} \sim \mathcal{G}(e, \eta_{k,t}), \quad \eta_{k,t} \sim \mathcal{G}(f, \nu) \tag{9}$$

$$\nu \sim \mathcal{G}(1/2, \zeta), \quad \zeta \sim \mathcal{G}(1/2, h^2) \tag{10}$$

### 1.3 Gibbs Sampling Updates

Denote $\boldsymbol{\Theta} = \{\boldsymbol{\Lambda}, \boldsymbol{S}, \boldsymbol{L}, \boldsymbol{U}, \boldsymbol{\phi}, \boldsymbol{\omega}, \boldsymbol{\rho}, \boldsymbol{\tau}, \boldsymbol{\eta}, \nu, \zeta\}$, we use Gibbs sampling to approximate the joint posterior distribution of $(\boldsymbol{Z}, \boldsymbol{\Theta})$,

1. Given $\boldsymbol{\Theta}$, find $p(z_{p,t}|\boldsymbol{\Theta}, \boldsymbol{Z} \in R(\boldsymbol{Y}), \boldsymbol{Z}_{-p,-t})$, for $p = 1, \ldots, P$, $t = 1, \ldots, T$.
2. Given $\boldsymbol{Z}$, find $p(\boldsymbol{\Theta}|\boldsymbol{Z}, \boldsymbol{Z} \in R(\boldsymbol{Y}))$ reduce to $p(\boldsymbol{\Theta}|\boldsymbol{Z})$

Treat $\boldsymbol{Z}$ as augmented data, the full likelihood for $(\boldsymbol{Z}, \boldsymbol{\Theta})$ is

$$
\begin{aligned}
p(\boldsymbol{Z}, \boldsymbol{\Theta}) = &\left(\prod_{t=1}^{T} \mathcal{N}(\boldsymbol{z}_t; \boldsymbol{\Lambda}\boldsymbol{s}_t, \boldsymbol{R})\right) \times \left(\prod_{k=1}^{K} \mathcal{G}(\phi_k; 1/2, \omega_k)\mathcal{G}(\omega_k; 1/2, d^2)\right) \\
&\times \prod_{k=1}^{K}\left[\mathcal{N}(s_{0,k}; 0, \sigma_s^2)\left(\prod_{t=1}^{T} \mathcal{N}(s_{k,t}; \rho_k s_{k,t-1}, \tau_{k,t})\mathcal{G}(\tau_{k,t}; e, \eta_{k,t})\mathcal{G}(\eta_{k,t}; f, \nu)\right)\right] \\
&\times \left(\prod_{p=1}^{P}\prod_{k=1}^{K} \mathcal{N}_+(\lambda_{p,k}; 0, l_{p,k})\mathcal{G}(l_{p,k}; a, u_{p,k})\mathcal{G}(u_{p,k}; b, \phi_k)\right) \times \prod_{k=1}^{K} \text{TN}_{(0,1)}(\rho_k; \mu_0, \sigma_0^2) \\
&\times \mathcal{G}(\nu; 1/2, \zeta) \times \mathcal{G}(\zeta; 1/2, h^2)
\end{aligned}
\tag{11}
$$

---

[*]contributed equally

- Sampling $z_{p,t}$

$$p(z_{p,t}|\boldsymbol{\Theta}, \boldsymbol{Z} \in R(\boldsymbol{Y}), \boldsymbol{Z}_{-p,-t}) \sim \text{TN}_{[\underline{z_{p,t}}, \overline{z_{p,t}}]}(\sum_{k=1}^{K} \lambda_{p,k} s_{k,t}, 1) \qquad (12)$$

where $\underline{z_{p,t}} = \max\{z_{p',t'} : y_{p',t'} < y_{p,t}\}$ and $\overline{z_{p,t}} = \min\{z_{p',t'} : y_{p',t'} > y_{p,t}\}$

- Sampling $\lambda_{p,k}$

$$p(\lambda_{p,k}|-) \propto \left( \prod_{t=1}^{T} \mathcal{N}(z_{p,t}; \lambda_{p,k} s_{k,t} + \sum_{j \neq k} s_{j,t} \lambda_{p,j}, 1) \right) \mathcal{N}_{+}(\lambda_{p,k}; 0, l_{p,k})$$

$$= \mathcal{N}_{+}\left( \lambda_{p,k}; v_{\lambda_{p,k}} \sum_{t=1}^{T} \left[ s_{k,t} z_{p,t} - s_{k,t} \sum_{j \neq k} s_{j,t} \lambda_{p,j} \right], v_{\lambda_{p,k}} \right)$$

$$v_{\lambda_{p,k}} = (l_{p,k}^{-1} + \sum_{t=1}^{T} s_{k,t}^2)^{-1} \qquad (13)$$

- Sampling $l_{p,k}$, $u_{p,k}$

$$p(l_{p,k}|-) = \text{GIG}(a - 1/2, 2u_{p,k}, (\lambda_{p,k})^2), \quad p(u_{p,k}|-) = \mathcal{G}(a + b, u_{p,k} + \phi_k) \qquad (14)$$

The Generalized Inverse Gaussian (GIG) distribution can be expressed as

$$\text{GIG}(x; p, a, b) = \frac{(a/b)^{\frac{p}{2}}}{2K_p(\sqrt{ab})} x^{P-1} \exp\left( -\frac{1}{2}(ax + \frac{b}{x}) \right) \quad (x > 0)$$

where $K_p(\theta)$ is the modified Bessel function of the second kind

$$K_p(\theta) = \int_0^{\infty} \frac{1}{2} \theta^{-p} t^{P-1} \exp\left( -\frac{1}{2}(t + \frac{\theta^2}{t}) \right) \, \mathsf{d}t$$

with property $K_{-\frac{1}{2}}(\theta) = \frac{1}{2}\sqrt{2\pi}\theta^{-\frac{1}{2}} \exp(-\theta)$ and $K_{p+1}(\theta) = K_{P-1}(\theta) + \frac{2p}{\theta} K_p(\theta)$ [1].

- Sampling $\phi_k$, $\omega_k$

$$p(\phi_k|-) = \mathcal{G}(1/2 + bP, \omega_k + \sum_{p=1}^{P} u_{p,k}), \quad p(\omega_k|-) = \mathcal{G}(1, \phi_k + d^2) \qquad (15)$$

- Sampling $\tau_{k,t}$, $\eta_{k,t}$

$$p(\tau_{k,t}|-) = \text{GIG}(e - 1/2, 2\eta_{k,t}, (s_{k,t} - \rho_k s_{k,t-1})^2), \quad p(\eta_{k,t}|-) = \mathcal{G}(e + f, \tau_{k,t} + \nu) \qquad (16)$$

- Sampling $\nu$, $\zeta$

$$p(\nu|-) = \mathcal{G}(1/2 + fTK, \zeta + \sum_{k=1}^{K} \sum_{t=1}^{T} \eta_{k,t}), \quad p(\zeta|-) = \mathcal{G}(1, \nu + h^2) \qquad (17)$$

- Sampling $\rho_k$

$$p(\rho_k|-) = \text{TN}_{(0,1)}\left( \sigma_{\rho_k}^2 (\sigma_0^{-2}\mu_0 + \sum_{t=1}^{T} \tau_{k,t}^{-1} s_{k,t-1} s_{k,t}), \sigma_{\rho_k}^2 \right) \qquad (18)$$

where $\sigma_{\rho_k}^2 = 1/(\sigma_0^{-2} + \sum_{t=1}^{T} \tau_{k,t}^{-1} s_{k,t-1}^2)$.

- Sampling $s_{k,t}$: we have the state model and the observation model[2]

$$\boldsymbol{s}_t|\boldsymbol{s}_{t-1} \sim \mathcal{N}(\boldsymbol{A}\boldsymbol{s}_{t-1}, \boldsymbol{Q}_t), \quad \boldsymbol{A} = \text{diag}(\boldsymbol{\rho}), \quad \boldsymbol{Q}_t = \text{diag}(\boldsymbol{\tau}_t), \qquad (19)$$

$$\boldsymbol{z}_t|\boldsymbol{s}_t \sim \mathcal{N}(\boldsymbol{\Lambda}\boldsymbol{s}_t, \boldsymbol{R}), \quad \boldsymbol{R} = \boldsymbol{I}_P \qquad (20)$$

for $t = 1, \ldots, T$

1. Forward Filtering: beginning at $t = 0$ with $\boldsymbol{s}_0 \sim \mathcal{N}(\boldsymbol{0}, \sigma_s^2 \boldsymbol{I}_K)$, we have, for all $t = 1, \dots, T$, the on-line posteriors $p(\boldsymbol{s}_t | \boldsymbol{z}_{1:t}) = \mathcal{N}(\boldsymbol{m}_t, \boldsymbol{V}_t)$. Start from

$$p(\boldsymbol{s}_{t-1} | \boldsymbol{z}_{1:(t-1)}) = \mathcal{N}(\boldsymbol{m}_{t-1}, \boldsymbol{V}_{t-1}) \tag{21}$$

Combine (19) with (21), integrate out $\boldsymbol{s}_{t-1}$, we have the predictive density at $t$,

$$p(\boldsymbol{s}_t | \boldsymbol{z}_{1:(t-1)}) = \mathcal{N}(\boldsymbol{A}\boldsymbol{m}_{t-1}, \boldsymbol{Q}_t + \boldsymbol{A}\boldsymbol{V}_{t-1}\boldsymbol{A}^T) \tag{22}$$

Further combine (20) with (22), we have the on-line posteriors at $t$, $p(\boldsymbol{s}_t | \boldsymbol{z}_{1:t}) = \mathcal{N}(\boldsymbol{m}_t, \boldsymbol{V}_t)$, where $\boldsymbol{m}_t = \boldsymbol{V}_t \{\boldsymbol{\Lambda}^T \boldsymbol{R}^{-1} \boldsymbol{z}_t + \boldsymbol{H}_t^{-1} \boldsymbol{A}\boldsymbol{m}_{t-1}\}$, $\boldsymbol{V}_t = [\boldsymbol{H}_t^{-1} + \boldsymbol{\Lambda}^T \boldsymbol{R}^{-1} \boldsymbol{\Lambda}]^{-1}$, and $\boldsymbol{H}_t = \boldsymbol{Q}_t + \boldsymbol{A}\boldsymbol{V}_{t-1}\boldsymbol{A}^T$.

2. Backward Sampling: define the backward smoothing density

$$p(\boldsymbol{s}_t | \boldsymbol{z}_{1:T}) = \mathcal{N}(\widetilde{\boldsymbol{m}}_t, \widetilde{\boldsymbol{V}}_t) \tag{23}$$

At $t = T$, we have the initialization condition $p(\boldsymbol{s}_T | \boldsymbol{z}_{1:T}) = \mathcal{N}(\widetilde{\boldsymbol{m}}_T, \widetilde{\boldsymbol{V}}_T) = \mathcal{N}(\boldsymbol{m}_T, \boldsymbol{V}_T)$. Combine (19) with (21), we have the conditional distribution of $\boldsymbol{s}_{t-1}$ given $\boldsymbol{s}_t$,

$$p(\boldsymbol{s}_{t-1} | \boldsymbol{s}_t, \boldsymbol{z}_{1:(t-1)}) = \mathcal{N}(\widetilde{\boldsymbol{\mu}}_{t-1}, \widetilde{\boldsymbol{\Sigma}}_{t-1}) \tag{24}$$

where $\widetilde{\boldsymbol{\mu}}_{t-1} = \widetilde{\boldsymbol{\Sigma}}_{t-1}\{\boldsymbol{A}^T \boldsymbol{Q}_t^{-1} \boldsymbol{s}_t + \boldsymbol{V}_{t-1}^{-1} \boldsymbol{m}_{t-1}\}$, $\widetilde{\boldsymbol{\Sigma}}_{t-1} = (\boldsymbol{V}_{t-1}^{-1} + \boldsymbol{A}^T \boldsymbol{Q}_t^{-1} \boldsymbol{A})^{-1}$.

3. Backward recursion for the posterior covariance: For each $t = T-1, T-2, \dots, 0$, start from (23), we are able to find $p(\boldsymbol{s}_{t-1} | \boldsymbol{z}_{1:T}) = \mathcal{N}(\widetilde{\boldsymbol{m}}_{t-1}, \widetilde{\boldsymbol{V}}_{t-1})$ via backward recursion. According to the Markov property,

$$p(\boldsymbol{s}_{t-1} | \boldsymbol{s}_{t:T}, \boldsymbol{z}_{1:T}) \equiv p(\boldsymbol{s}_{t-1} | \boldsymbol{s}_t, \boldsymbol{z}_{1:T}) \equiv p(\boldsymbol{s}_{t-1} | \boldsymbol{s}_t, \boldsymbol{z}_{1:(t-1)}) \tag{25}$$

using the second equality in (25), we obtain

$$p(\boldsymbol{s}_{t-1} | \boldsymbol{s}_t, \boldsymbol{z}_{1:T}) = \mathcal{N}(\widetilde{\boldsymbol{\mu}}_{t-1}, \widetilde{\boldsymbol{\Sigma}}_{t-1}) \tag{26}$$

Combine (23) with (26), integrated out $\boldsymbol{s}_t$, we have the backward smoothing density at $t-1$,

$$p(\boldsymbol{s}_{t-1} | \boldsymbol{z}_{1:T}) = \mathcal{N}(\widetilde{\boldsymbol{m}}_{t-1}, \widetilde{\boldsymbol{V}}_{t-1})$$
$$\widetilde{\boldsymbol{m}}_{t-1} = \widetilde{\boldsymbol{\Sigma}}_{t-1}(\boldsymbol{A}^T \boldsymbol{Q}_t^{-1} \widetilde{\boldsymbol{m}}_t + \boldsymbol{V}_{t-1}^{-1} \boldsymbol{m}_{t-1})$$
$$\widetilde{\boldsymbol{V}}_{t-1} = \widetilde{\boldsymbol{\Sigma}}_{t-1} + \widetilde{\boldsymbol{\Sigma}}_{t-1} \boldsymbol{A}^T \boldsymbol{Q}_t^{-1} \widetilde{\boldsymbol{V}}_t \boldsymbol{Q}_t^{-1} \boldsymbol{A} \widetilde{\boldsymbol{\Sigma}}_{t-1}$$
$$\tag{27}$$

## 2   Gibbs Sampling for the Extended Model

### 2.1   Dealing with Multiple Documents

At each time point $t$, for each document $n_t$, the likelihood is

$$y_{p,t}^{n_t} = g\left(z_{p,t}^{n_t}\right) \tag{28}$$

To consider $N_t$ documents per time point, add additional layer,

$$\boldsymbol{z}_t^{n_t} = \boldsymbol{\Lambda} \boldsymbol{b}_t^{n_t} + \boldsymbol{\epsilon}_t^{n_t}, \quad \boldsymbol{\epsilon}_t^{n_t} \sim \mathcal{N}(\boldsymbol{0}, \boldsymbol{R}), \quad \boldsymbol{R} = \boldsymbol{I}_P, \quad n_t = 1, \dots, N_t$$
$$\boldsymbol{b}_t^{n_t} \sim \mathcal{N}(\boldsymbol{s}_t, \boldsymbol{\Gamma}), \quad \boldsymbol{\Gamma} = \mathrm{diag}(\boldsymbol{\gamma}), \quad \gamma_k^{-1} \sim \mathcal{G}(\alpha, \beta), \quad k = 1, \dots, K \tag{29}$$

### 2.2   Gibbs Sampler

Denote $\boldsymbol{\Theta} = \{\boldsymbol{\Lambda}, \boldsymbol{S}, \boldsymbol{B}_{1:T}, \boldsymbol{L}, \boldsymbol{U}, \boldsymbol{\Gamma}, \boldsymbol{\phi}, \boldsymbol{\omega}, \boldsymbol{\rho}, \boldsymbol{\tau}, \boldsymbol{\eta}, \nu, \zeta\}$, we use Gibbs sampling to approximate the joint posterior distribution of $(\boldsymbol{Z}, \boldsymbol{\Theta})$,

1. Given $\boldsymbol{\Theta}$, find $p(z_{p,t}^{n_t} | \boldsymbol{\Theta}, \boldsymbol{Z} \in R(\boldsymbol{Y}), \boldsymbol{Z}_{-p,-t,-n_t})$, for $p = 1, \dots, P$, $t = 1, \dots, T$, $n_t = 1, \dots, N_t$

2. Given $\boldsymbol{Z}$, find $p(\boldsymbol{\Theta} | \boldsymbol{Z}, \boldsymbol{Z} \in R(\boldsymbol{Y}))$ reduce to $p(\boldsymbol{\Theta} | \boldsymbol{Z})$

Treat $\boldsymbol{Z}$ as augmented data, the full likelihood for $(\boldsymbol{Z}, \boldsymbol{\Theta})$ is

$$
\begin{aligned}
p(\boldsymbol{Z}, \boldsymbol{\Theta}) = & \left( \prod_{t=1}^{T} \prod_{n_t=1}^{N_t} \mathcal{N}(\boldsymbol{z}_t^{n_t}; \boldsymbol{\Lambda} \boldsymbol{b}_t^{n_t}, \boldsymbol{R}) \right) \times \left( \prod_{k=1}^{K} \mathcal{G}(\phi_k; 1/2, \omega_k) \mathcal{G}(\omega_k; 1/2, d^2) \right) \\
& \times \left( \prod_{t=1}^{T} \prod_{n_t=1}^{N_t} \mathcal{N}(\boldsymbol{b}_t^{n_t}, \boldsymbol{s}_t, \boldsymbol{\Gamma}) \right) \times \prod_{k=1}^{K} \mathcal{G}(\gamma_k^{-1}; \alpha, \beta) \\
& \times \prod_{k=1}^{K} \left[ \mathcal{N}(s_{0,k}; 0, \sigma_s^2) \left( \prod_{t=1}^{T} \mathcal{N}(s_{k,t}; \rho_k s_{k,t-1}, \tau_{k,t}) \mathcal{G}(\tau_{k,t}; e, \eta_{k,t}) \mathcal{G}(\eta_{k,t}; f, \nu) \right) \right] \\
& \times \left( \prod_{p=1}^{P} \prod_{k=1}^{K} \mathcal{N}_+(\lambda_{p,k}; 0, l_{p,k}) \mathcal{G}(l_{p,k}; a, u_{p,k}) \mathcal{G}(u_{p,k}; b, \phi_k) \right) \times \prod_{k=1}^{K} \mathrm{TN}_{(0,1)}(\rho_k; \mu_0, \sigma_0^2) \\
& \times \mathcal{G}(\nu; 1/2, \zeta) \times \mathcal{G}(\zeta; 1/2, h^2) \qquad (30)
\end{aligned}
$$

- Sampling $z_{p,t}^{n_t}$

$$
p(z_{p,t}^{n_t} | \boldsymbol{\Theta}, \boldsymbol{Z} \in R(\boldsymbol{Y}), \boldsymbol{Z}_{-p,-t,-n_t}) \sim \mathrm{TN}_{[\underline{z_{p,t}^{n_t}}, \overline{z_{p,t}^{n_t}}]} (\sum_{k=1}^{K} \lambda_{p,k} b_{k,t}^{n_t}, 1) \qquad (31)
$$

where $\underline{z_{p,t}^{n_t}} = \max\{z_{p',t'}^{n_t} : y_{p',t'}^{n_t} < y_{p,t}^{n_t}\}$ and $\overline{z_{p,t}^{n_t}} = \min\{z_{p',t'}^{n_t} : y_{p',t'}^{n_t} > y_{p,t}^{n_t}\}$

- Sampling $\lambda_{p,k}$

$$
\begin{aligned}
p(\lambda_{p,k} | -) & \propto \left( \prod_{t=1}^{T} \prod_{n_t=1}^{N_t} \mathcal{N}(z_{p,t}^{n_t}; \lambda_{p,k} b_{k,t}^{n_t} + \sum_{j \neq k} b_{j,t}^{n_t} \lambda_{p,j}, 1) \right) \mathcal{N}_+(\lambda_{p,k}; 0, l_{p,k}) \\
& = \mathcal{N}_+ \left( \lambda_{p,k}; v_{\lambda_{p,k}} \sum_{t=1}^{T} \sum_{n_t=1}^{N_t} b_{k,t}^{n_t} \left[ z_{p,t}^{n_t} - \boldsymbol{\lambda}_p \boldsymbol{b}_t^{n_t} + b_{k,t}^{n_t} \lambda_{p,k} \right], v_{\lambda_{p,k}} \right) \\
v_{\lambda_{p,k}} & = (l_{p,k}^{-1} + \sum_{t=1}^{T} \sum_{n_t=1}^{N_t} (b_{k,t}^{n_t})^2)^{-1} \qquad (32)
\end{aligned}
$$

- Sampling $\boldsymbol{b}_t^{n_t}$

$$
p(\boldsymbol{b}_t^{n_t} | -) = \mathcal{N}(\boldsymbol{\Sigma}_{\boldsymbol{b}_t^{n_t}}(\boldsymbol{\Lambda}^T \boldsymbol{R}^{-1} \boldsymbol{z}_t^{n_t} + \boldsymbol{\Gamma}^{-1} \boldsymbol{s}_t), \boldsymbol{\Sigma}_{\boldsymbol{b}_t^{n_t}}), \quad \boldsymbol{\Sigma}_{\boldsymbol{b}_t^{n_t}} = (\boldsymbol{\Gamma}^{-1} + \boldsymbol{\Lambda}^T \boldsymbol{R}^{-1} \boldsymbol{\Lambda})^{-1} \qquad (33)
$$

- Sampling $\gamma_k^{-1}$

$$
p(\gamma_k^{-1} | -) \sim \mathcal{G} \left( \alpha + \frac{1}{2} \sum_{t=1}^{T} N_t, \beta + \frac{1}{2} \sum_{t=1}^{T} \sum_{n_t=1}^{N_t} (b_{k,t}^{n_t} - s_{k,t})^2 \right) \qquad (34)
$$

- Sampling $l_{p,k}, u_{p,k}$

$$
p(l_{p,k} | -) = \mathrm{GIG}(a - 1/2, 2u_{p,k}, (\lambda_{p,k})^2), \quad p(u_{p,k} | -) = \mathcal{G}(a + b, l_{p,k} + \phi_k) \qquad (35)
$$

- Sampling $\phi_k, \omega_k$

$$
p(\phi_k | -) = \mathcal{G}(1/2 + bP, \omega_k + \sum_{p=1}^{P} u_{p,k}), \quad p(\omega_k | -) = \mathcal{G}(1, \phi_k + d^2) \qquad (36)
$$

- Sampling $\tau_{k,t}, \eta_{k,t}$

$$
p(\tau_{k,t} | -) = \mathrm{GIG}(e - 1/2, 2\eta_{k,t}, (s_{k,t} - \rho_k s_{k,t-1})^2), \quad p(\eta_{k,t} | -) = \mathcal{G}(e + f, \tau_{k,t} + \nu) \qquad (37)
$$

- Sampling $\nu, \zeta$

$$
p(\nu | -) = \mathcal{G}(1/2 + fTK, \zeta + \sum_{k=1}^{K} \sum_{t=1}^{T} \eta_{k,t}), \quad p(\zeta | -) = \mathcal{G}(1, \nu + h^2) \qquad (38)
$$

- Sampling $\rho_k$

$$p(\rho_k|-) = \mathrm{TN}_{(0,1)}\left(\sigma_{\rho_k}^2(\sigma_0^{-2}\mu_0 + \sum_{t=1}^{T}\tau_{k,t}^{-1}s_{k,t-1}s_{k,t}), \sigma_{\rho_k}^2\right) \tag{39}$$

where $\sigma_{\rho_k}^2 = 1/(\sigma_0^{-2} + \sum_{t=1}^{T}\tau_{k,t}^{-1}s_{k,t-1}^2)$.

We have the state model and the observation model[3]

$$\boldsymbol{s}_t|\boldsymbol{s}_{t-1} \sim \mathcal{N}(\boldsymbol{A}\boldsymbol{s}_{t-1}, \boldsymbol{Q}_t), \quad \boldsymbol{A} = \mathrm{diag}(\boldsymbol{\rho}), \quad \boldsymbol{Q}_t = \mathrm{diag}(\boldsymbol{\tau}_t), \tag{40}$$

$$\boldsymbol{b}_t^{n_t}|\boldsymbol{s}_t \sim \mathcal{N}(\boldsymbol{s}_t, \boldsymbol{\Gamma}), \quad \boldsymbol{\Gamma} = \mathrm{diag}(\gamma_k) \tag{41}$$

for $n_t = 1, \ldots, N_t, t = 1, \ldots, T$

1. Forward Filtering: beginning at $t = 0$ with $\boldsymbol{s}_0 \sim \mathcal{N}(\boldsymbol{0}, \sigma_s^2\boldsymbol{I}_K)$, we have, for all $t = 1, \ldots, T$, the on-line posteriors $p(\boldsymbol{s}_t|\boldsymbol{B}_{1:t}) = \mathcal{N}(\boldsymbol{m}_t, \boldsymbol{V}_t)$. Start from

$$p(\boldsymbol{s}_{t-1}|\boldsymbol{B}_{1:(t-1)}) = \mathcal{N}(\boldsymbol{m}_{t-1}, \boldsymbol{V}_{t-1}) \tag{42}$$

Combine (40) with (42), integrate out $\boldsymbol{s}_{t-1}$, we have the predictive density at $t$,

$$p(\boldsymbol{s}_t|\boldsymbol{B}_{1:(t-1)}) = \mathcal{N}(\boldsymbol{A}\boldsymbol{m}_{t-1}, \boldsymbol{Q}_t + \boldsymbol{A}\boldsymbol{V}_{t-1}\boldsymbol{A}^T) \tag{43}$$

Further combine (41) with (43), we have the on-line posteriors at $t$,

$$p(\boldsymbol{s}_t|\boldsymbol{B}_{1:t}) = \mathcal{N}(\boldsymbol{m}_t, \boldsymbol{V}_t)$$
$$\boldsymbol{m}_t = \boldsymbol{V}_t\{N_t\boldsymbol{\Gamma}^{-1}\overline{\boldsymbol{b}_t} + (\boldsymbol{Q}_t + \boldsymbol{A}\boldsymbol{V}_{t-1}\boldsymbol{A}^T)^{-1}\boldsymbol{A}\boldsymbol{m}_{t-1}\},$$
$$\boldsymbol{V}_t = [(\boldsymbol{Q}_t + \boldsymbol{A}\boldsymbol{V}_{t-1}\boldsymbol{A}^T)^{-1} + N_t\boldsymbol{\Gamma}^{-1}]^{-1}, \quad \overline{\boldsymbol{b}_t} = \frac{1}{N_t}\sum_{n_t=1}^{N_t}\boldsymbol{b}_t^{n_t} \tag{44}$$

Define $\widetilde{\boldsymbol{\Omega}}_t = (\boldsymbol{Q}_t + \boldsymbol{A}\boldsymbol{V}_{t-1}\boldsymbol{A}^T)$, according to the Woodbury lemma,

$$(\boldsymbol{A} + \boldsymbol{U}\boldsymbol{C}\boldsymbol{V})^{-1} = \boldsymbol{A}^{-1} - \boldsymbol{A}^{-1}\boldsymbol{U}(\boldsymbol{C}^{-1} + \boldsymbol{V}\boldsymbol{A}^{-1}\boldsymbol{U})^{-1}\boldsymbol{V}\boldsymbol{A}^{-1} \tag{45}$$

we have

$$(\widetilde{\boldsymbol{\Omega}}_t^{-1} + N_t\boldsymbol{\Gamma}^{-1})^{-1} = \widetilde{\boldsymbol{\Omega}}_t - \widetilde{\boldsymbol{\Omega}}_t(N_t^{-1}\boldsymbol{\Gamma} + \widetilde{\boldsymbol{\Omega}}_t)^{-1}\widetilde{\boldsymbol{\Omega}}_t \tag{46}$$

2. Backward Sampling: define the backward smoothing density

$$p(\boldsymbol{s}_t|\boldsymbol{B}_{1:T}) = \mathcal{N}(\widetilde{\boldsymbol{m}}_t, \widetilde{\boldsymbol{V}}_t) \tag{47}$$

At $t = T$, we have the initialization condition $p(\boldsymbol{s}_T|\boldsymbol{B}_{1:T}) = \mathcal{N}(\widetilde{\boldsymbol{m}}_T, \widetilde{\boldsymbol{V}}_T) = \mathcal{N}(\boldsymbol{m}_T, \boldsymbol{V}_T)$. Combine (40) with (42), we have the conditional distribution of $\boldsymbol{s}_{t-1}$ given $\boldsymbol{s}_t$,

$$p(\boldsymbol{s}_{t-1}|\boldsymbol{s}_t, \boldsymbol{B}_{1:(t-1)}) = \mathcal{N}(\widetilde{\boldsymbol{\mu}}_{t-1}, \widetilde{\boldsymbol{\Sigma}}_{t-1})$$
$$\widetilde{\boldsymbol{\mu}}_{t-1} = \widetilde{\boldsymbol{\Sigma}}_{t-1}\{\boldsymbol{A}^T\boldsymbol{Q}_t^{-1}\boldsymbol{s}_t + \boldsymbol{V}_{t-1}^{-1}\boldsymbol{m}_{t-1}\}$$
$$\widetilde{\boldsymbol{\Sigma}}_{t-1} = (\boldsymbol{V}_{t-1}^{-1} + \boldsymbol{A}^T\boldsymbol{Q}_t^{-1}\boldsymbol{A})^{-1} \tag{48}$$

Similarly, apply Woodbury matrix inversion lemma we have

$$\widetilde{\boldsymbol{\Sigma}}_{t-1} = \boldsymbol{V}_{t-1} - \boldsymbol{V}_{t-1}\boldsymbol{A}^T\widetilde{\boldsymbol{\Omega}}_t^{-1}\boldsymbol{A}\boldsymbol{V}_{t-1} \tag{49}$$

- Sampling $\widetilde{\boldsymbol{V}}_{0:(T-1)}$

  Integrated out $\boldsymbol{B}_t$, we have the observation model

$$\boldsymbol{z}_t^{n_t}|\boldsymbol{s}_t \sim \mathcal{N}(\boldsymbol{\Lambda}\boldsymbol{s}_t, \widetilde{\boldsymbol{R}}), \quad \widetilde{\boldsymbol{R}} = \boldsymbol{I}_P + \boldsymbol{\Lambda}\boldsymbol{\Gamma}\boldsymbol{\Lambda}^T, \quad \widetilde{\boldsymbol{R}}^{-1} = \boldsymbol{I}_P - \boldsymbol{\Lambda}(\boldsymbol{\Gamma}^{-1} + \boldsymbol{\Lambda}^T\boldsymbol{\Lambda})^T\boldsymbol{\Lambda}^T \tag{50}$$

for $n_t = 1, \ldots, N_t, t = 1, \ldots, T$. We have the on-line posteriors at $t$,

$$p(\boldsymbol{s}_t|\boldsymbol{Z}_{1:t}) = \mathcal{N}(\boldsymbol{m}_t, \boldsymbol{V}_t)$$
$$\boldsymbol{m}_t = \boldsymbol{V}_t\{N_t\boldsymbol{\Lambda}^T\widetilde{\boldsymbol{R}}^{-1}\overline{\boldsymbol{z}}_t + \widetilde{\boldsymbol{\Omega}}_t^{-1}\boldsymbol{A}\boldsymbol{m}_{t-1}\},$$
$$\boldsymbol{V}_t = [\widetilde{\boldsymbol{\Omega}}_t^{-1} + N_t\boldsymbol{\Lambda}^T\widetilde{\boldsymbol{R}}^{-1}\boldsymbol{\Lambda}]^{-1}, \quad \overline{\boldsymbol{z}}_t = \frac{1}{N_t}\sum_{n_t=1}^{N_t}\boldsymbol{z}_t^{n_t} \tag{51}$$

The conditional distribution of $\boldsymbol{s}_{t-1}$ given $\boldsymbol{s}_t$,

$$p(\boldsymbol{s}_{t-1}|\boldsymbol{s}_t, \boldsymbol{Z}_{1:(t-1)}) = \mathcal{N}(\widetilde{\boldsymbol{\mu}}_{t-1}, \widetilde{\boldsymbol{\Sigma}}_{t-1})$$

$$\widetilde{\boldsymbol{\mu}}_{t-1} = \widetilde{\boldsymbol{\Sigma}}_{t-1}\{\boldsymbol{A}^T\boldsymbol{Q}_t^{-1}\boldsymbol{s}_t + \boldsymbol{V}_{t-1}^{-1}\boldsymbol{m}_{t-1}\}$$

$$\widetilde{\boldsymbol{\Sigma}}_{t-1} = (\boldsymbol{V}_{t-1}^{-1} + \boldsymbol{A}^T\boldsymbol{Q}_t^{-1}\boldsymbol{A})^{-1} \tag{52}$$

Similarly, apply Woodbury matrix inversion lemma we have

$$\widetilde{\boldsymbol{\Sigma}}_{t-1} = \boldsymbol{V}_{t-1} - \boldsymbol{V}_{t-1}\boldsymbol{A}^T\widetilde{\boldsymbol{\Omega}}_t^{-1}\boldsymbol{A}\boldsymbol{V}_{t-1} \tag{53}$$

Further, the backward smoothing density at $t-1$,

$$p(\boldsymbol{s}_{t-1}|\boldsymbol{Z}_{1:T}) = \mathcal{N}(\widetilde{\boldsymbol{m}}_{t-1}, \widetilde{\boldsymbol{V}}_{t-1})$$

$$\widetilde{\boldsymbol{m}}_{t-1} = \widetilde{\boldsymbol{\Sigma}}_{t-1}(\boldsymbol{A}^T\boldsymbol{Q}_t^{-1}\widetilde{\boldsymbol{m}}_t + \boldsymbol{V}_{t-1}^{-1}\boldsymbol{m}_{t-1})$$

$$\widetilde{\boldsymbol{V}}_{t-1} = \widetilde{\boldsymbol{\Sigma}}_{t-1} + \widetilde{\boldsymbol{\Sigma}}_{t-1}\boldsymbol{A}^T\boldsymbol{Q}_t^{-1}\widetilde{\boldsymbol{V}}_t\boldsymbol{Q}_t^{-1}\boldsymbol{A}\widetilde{\boldsymbol{\Sigma}}_{t-1}$$

$$\tag{54}$$

## 3  Experimental results

### 3.1  Simulation Study: DRFM with different innovations

We conducted a simulation study to assess the performance of our proposed approach. We first generate the latent continuous variable $\boldsymbol{Z}$ from the augmented model with $e = f = 0.5$, $\nu = 1$, $\rho_k = 0.5$, $l_{p,k} = 1/P$ for $k = 1, \ldots, K$, $p = 1, \ldots, P$ and then round it to integer value. Three different approaches are considered here: Gaussian innovation with fixed variance $\delta \sim \mathcal{N}(0,1)$, Gaussian innovation with unknown variance $\delta \sim \mathcal{N}(0,\tau)$, and $\tau^{-1} \sim \mathcal{G}(0.01, 0.01)$, and heavy-tailed innovation $\delta \sim \text{TPBN}(0.5, 0.5, \phi)$ with $\phi^{1/2} \sim \mathcal{C}^+(0,1)$. The results are shown in Figure 1.

Figure 1: Estimated posterior mean of factor score $s$ with $95\%$ confidence interval for $P = 10$, $K = 2$ and $T = 150$. Left column: Gaussian innovation with fixed variance $\tau = 1$; middle column: Gaussian innovation with unknown variance $\tau^{-1} \sim \mathcal{G}(0.01, 0.01)$; right column: heavy-tailed innovation.

Note that here we are dealing with a simple two factor dynamic model, and we are lucky to recover the ground truth of the trajectory of factor score. In contrast to other models that involve nonlinear transformation, the smooth transition and sudden jumps can be well preserved under the proposed DFRM framework, using heavy-tailed innovation.

Figure 2 shows the monotone relationship between observation $\boldsymbol{y}$ and the latent variable $\boldsymbol{z}$ inferred by extended rank likelihood in our DFRM model. It can be seen that the rank likelihood approach maintains the order information of $\boldsymbol{y}$ in $\boldsymbol{z}$ and provides a flexible link between $\boldsymbol{y}$ and $\boldsymbol{z}$.

Figure 2: Estimated posterior mean of latent variable $z$ vs. the observed data $y$ for $P = 10$, $K = 2$ and $T = 150$. Left: Gaussian innovation with fixed variance $\tau = 1$; middle: Gaussian innovation with unknown variance $\tau^{-1} \sim \mathcal{G}(0.01, 0.01)$; right: heavy-tailed innovation.

## 3.2 Case Study I: State of the Union dataset

The State of the Union dataset contains the transcripts of 225 US State of the Union addresses, from 1790 to 2014. We take each transcript as a document, $i.e.$, there is one document per year. We have 7518 unique words in total. Table 1 shows all 25 learned topics and and the top 12 most probable words associated with each of them. Figure 3 presents the learned trajectory for each topic.

Table 1: Top 12 words associated with the State of the Union Topics

| Topic#1 | Topic#2 | Topic#3 | Topic#4 | Topic#5 | Topic#6 | Topic#7 |
|---|---|---|---|---|---|---|
| UNITED | Dollars | ADMINISTRATION | GOVERNMENT | GOVERNMENT | LAW | GOVERNMENT |
| ACT | WAR | FEDERAL | AMERICAN | SERVICE | COUNTRY | UNITED |
| PUBLIC | MILLION | PROGRAM | UNITED | PUBLIC | NATIONAL | DEPARTMENT |
| TREATY | FISCAL | POLICY | FOREIGN | DEPARTMENT | PUBLIC | PUBLIC |
| Duties | EXPENDITURES | ENERGY | DEPARTMENT | Report | BUSINESS | LAW |
| Present | GOVERNMENT | Programs | NATIONAL | SECRETARY | GOVERNMENT | COURT |
| NATIONS | Billion | ECONOMIC | CANAL | DISTRICT | ACTION | SERVICE |
| TREASURY | PROGRAM | DEVELOPMENT | POLICY | Attention | CONTROL | FEDERAL |
| Session | UNITED | SECURITY | REPUBLIC | Present | UNITED | CANAL |
| COMMERCE | FEDERAL | Nation | Order | FISCAL | INTERSTATE | TARIFF |
| Citizens | Estimated | Major | ADMINISTRATION | LAWS | LABOR | DISTRICT |
| WAR | LEGISLATION | ACT | BANKS | COURT | CORPORATIONS | LANDS |

| Topic#8 | Topic#9 | Topic#10 | Topic#11 | Topic#12 | Topic#13 |
|---|---|---|---|---|---|
| GOVERNMENT | Constitution | MEXICO | INCREASE | GOVERNMENT | GOVERNMENT |
| GENERAL | COUNTRY | GOVERNMENT | UNITED | PUBLIC | UNITED |
| PUBLIC | WAR | Texas | Cent | Nation | ISLANDS |
| CHARACTER | PRESIDENT | UNITED | LAW | AMERICAN | COMMISSION |
| Interests | POWER | WAR | LEGISLATION | LAW | Island |
| Subject | MEXICO | MEXICAN | SECRETARY | POWER | Cuba |
| COUNTRY | PUBLIC | ARMY | Free | CONDITIONS | Spain |
| POWER | UNION | Territory | INCREASED | BUSINESS | ACT |
| Duty | California | COUNTRY | FISCAL | ISLANDS | GENERAL |
| Attention | SERVICE | PEACE | AMERICAN | SERVICE | MILITARY |
| FEDERAL | HOUSE | POLICY | TARIFF | WAR | INTERNATIONAL |
| Means | Period | LANDS | Products | LAWS | OFFICERS |

| Topic#14 | Topic#15 | Topic#16 | Topic#17 | Topic#18 | Topic#19 |
|---|---|---|---|---|---|
| Free | GOVERNMENT | STATUTE | Jobs | CHILDREN | AMERICA |
| NATIONS | FEDERAL | LAW | COUNTRY | AMERICA | GOVERNMENT |
| FREEDOM | PUBLIC | BUSINESS | TAX | AMERICANS | Nation |
| ECONOMIC | NATIONAL | GENERAL | AMERICAN | CARE | AMERICAN |
| MILITARY | COUNTRY | AMERICAN | ECONOMY | Tonight | FEDERAL |
| DEFENSE | ECONOMIC | PURPOSE | DEFICIT | Support | Tonight |
| UNITED | AGRICULTURE | COURT | AMERICANS | Century | PEACE |
| PEACE | BANKS | MEXICO | ENERGY | HEALTH | WAR |
| STRENGTH | Present | SERVICE | Businesses | Working | FREEDOM |
| SECURITY | AMERICA | FEDERAL | HEALTH | Challenge | AMERICANS |
| PROGRAM | REDUCTION | COMMISSION | PLAN | SECURITY | FUTURE |
| Nation | Construction | Present | CARE | Families | BUDGET |

| Topic#20 | Topic#21 | Topic#22 | Topic#23 | Topic#24 | Topic#25 |
|---|---|---|---|---|---|
| Gold | GOVERNMENT | WAR | GOVERNMENT | UNITED | GOVERNMENT |
| GOVERNMENT | Constitution | MEXICO | UNITED | TREATY | PUBLIC |
| Notes | UNITED | PEACE | Spain | Isthmus | BANKS |
| TREASURY | POWER | ARMY | Cuba | PUBLIC | BANK |
| Silver | UNION | ENEMY | Spanish | PANAMA | CURRENCY |
| UNITED | FEDERAL | FORCES | WAR | LAW | Money |
| Bonds | Duty | MILITARY | Island | Territory | UNITED |
| CURRENCY | AMERICAN | MEXICAN | SECRETARY | AMERICA | FEDERAL |
| RESERVE | Kansas | Production | June | CANAL | AMERICAN |
| Circulation | Question | JAPANESE | Duty | SERVICE | NATIONAL |
| Issued | LAW | FIGHTING | DEPARTMENT | BANKS | Duty |
| Large | Present | AMERICAN | FISCAL | Colombia | Institutions |

Figure 3: (***State of the Union* dataset**) Time evolving topics from 1790 to 2014. Left up panel: Topics 1 to 7. Right up panel: Topics 8 to 13. Left bottom panel: Topics 14 to 19. Right bottom panel: Topics 20 to 25. The plotted values represent the posterior means.

## Footnotes

[1] Code for simulating GIG distribution is available at: `http://jonaswallin.github.io/articles/2013/07/simulation-of-gig-distribution/`

[2] For brevity, we omit the dependencies on $\boldsymbol{\Theta}$ in notation

[3]For brevity, we omit the dependencies on $\boldsymbol{\Theta}$ in notation