[Reviews · NeurIPS 2014]

Submitted by Assigned_Reviewer_5

The paper introduces a theoretical framework which combines both dynamic and correlated topic models. The proposed approach is based on a latent factor model. The authors provide an interesting discussion on admixture models (traditional topic models) versus factor models. One of the main advantages of the chosen approach is the ability to model both positive topic usage and negative topic usage. Another advantage is the flexibility to model topical jumps over time. In addition an accelerated MCMC sampling procedure is provided.

The paper could be a landmark in the field of topic modeling - I very much hope to see it at the conference.

The paper is mathematically vigorous while being relatively understandable. (Although I hope the authors publish an extended version for the uninitiated.) The paper is well situated within the related work.

The evaluation is mostly annecdotal, with a case study of 225 State of the Union addresses, and a set of approx 140,000 scientific articles from JSTOR. On both data sets the topic model could identify important events and paradigm shifts.

The sole weaknesses of the paper is a lacking objective evaluation for a prediction problem with respect to available baselines. However, given the density of the material, it is forgivable to leave this for future work.

Summary: The paper could be a landmark in the field of topic modeling.

Submitted by Assigned_Reviewer_12

The authors propose a model for high-dimensional time series multivariate ordinal data (or data that can be treated as ordinal). The model uses the extended rank likelihood, a semi-parametric method, for robustness to model misspecification. Global latent "trend" parameters are given autoregressive dynamics, and three-parameter beta mixture of normal priors are used both to allow "jumps" in these trend parameters and to impose sparsity in the model's factor loadings. Applied to topic modeling, the proposed technique outputs a measure of the importance of different topics over time, as well as topic-topic and word-word correlations at each time point. The Gibbs sampling approach uses forward filtering backward sampling, and for scalability employs a subsampling scheme along the lines of Korattikara et al. (2014).

The proposed approach is novel and has many desirable properties for topic modeling in practice (sparsity, modeling topic prevalence over time, robustness, captures jumps in topics, outputs topic and word correlations for visualization purposes). It is also appreciated that the authors consider the scalability of the method, by using the subsampled MCMC approach. It should also be noted that the model is general purpose and could potentially be applied in domains other than topic modeling.

On the downside, the authors do not make any empirical comparison to the dynamic topic model or the correlated topic model, or even to standard LDA. The simulated data experiments in the supplementary material are useful however, demonstrating that the method can capture rapid jumps in the latent factor scores.

Another, simpler way to get correlation/similarity scores between topics (or words) is to take dot products of the columns of the topic matrix (or rows of the topic matrix, to get similarities between words). Such scores could be used to create visualizations similar to Figure 2. It wasn't clear from the manuscript what the benefit of the model-based correlation score is over this simple method.

**** Thanks to the authors for their response. It is a fair point about the nonparametric marginal likelihood vis-à-vis perplexity. I have updated my score accordingly.
Summary: The proposed model for time series topic modeling / multivariate ordinal data modeling has many practically useful properties. The main weakness of the paper is a lack of comparison to baseline methods, although this is somewhat mitigated in that the proposed technique is more powerful and informative in its output than previous models, reducing some of the burden of beating those models in terms of held-out prediction performance.

Submitted by Assigned_Reviewer_37

This paper proposes the dynamic rank factor models, which can extract topics changes along time, and also discover the topic and word correlations. The two case studies using the proposed model show interesting topic changing patterns, which are aligned well with true events; and the structures between topics and words are also visualized and discussed.

In general, this is a good paper. However, I have two main concerns.

First, while the introduction is well motivated and the experiment section is clearly written, Section 2 is not well structured and under selling.

- The authors start from dynamic factor models (section 2.1), and then talk about general topic models (without time involvement) and then come back to dynamic models again in section 2.3. This structure can be improved to avoid jump back and forth.

- Section 2.2 is not related with dynamic changing, but the authors use the subscript "t" to denote the t^th document, while using "t" to denote the time t in Section 2.1 and Section 2.3, and "n_t" to denote the document at time t. This is very confusing.

- Probably because the math part is dense, the proposed model in general is mixed with existing work and is not well emphasized. Even the dynamic part (Section 2.3), an important proposed work, only gives the latent factor model, but does not say anything about dynamic changing but refer back to Section 2.1

- It would be stronger if the nice properties of the proposed model are discussed more, instead of just listing.

Second, while the authors indicates that the proposed model is an alternative to dynamic topic models and correlated topic models, there is no comparison at all against these models in the experiment section.

- The experiment shows the changing of topics posterior mean along time and the detected topics, however, there is no evaluation of topic quality, like topic coherence, etc.
- This paper would be stronger with comparison against dynamic topic models and correlated topic models.
- For some interesting topics, it would be nice to see the actual topic changes (the top words) along time.

Some minor comments:

- \alpha and \beta are used as the parameters for gamma distribution and also the parameters for Dirichlet distributions.

- Figure 1, the displayed topics are the final topics after observing all documents? This should be clear.

- Figure 2, it would be nice to display a topic name rather than topic index, it would be easier to interprete the meanings of the topics.

- Figure 2, it's hard to read the word dendrogram, too small. It would be great if there is more discussion about the results.
Summary: This paper proposes an interesting rank factor model with interesting case studies. The model section can be better structured, and it would be stronger if there is more comparison and discussions about the results.
Author Feedback
Author rebuttal: We thank the reviewers for their careful reading of our manuscript, and for providing their thoughtful comments.

To Review 1:

Q1: On the downside, the authors do not make any empirical comparison to the dynamic topic model or the correlated topic model, or even to standard LDA.

A1: The proposed model is a semi-parametric methodology, which offers more model flexibilities, reduces the effect of model misspecification, and enjoys locally conjugate inference. However, the price we have to pay is that we lose the explicit form of the marginal likelihood (because it is nonparametric).

Since the marginal likelihood is distribution-free, it is difficult to make a direct quantitative comparison (e.g., perplexity) to models that assume a specific distribution form, like LDA and its variants. It was not that we sought to avoid such comparisons, rather, the challenge is coming up with a method of fair quantitative comparison, as our model and LDA-like models are fundamentally different. We are seeking objective evaluation criteria, which allow us to conduct fair comparison with other baselines in the literature.

Q2: Simpler way to get correlation/similarity scores between topics (or words) is to take dot products of the columns of the topic matrix, to get similarity between words. What the benefit of the model-based correlation score is over this simple method.

A2: A dot product of the vectors provides an empirical assessment of pairwise similarities. This method is admittedly simpler but less rigorous. The proposed method resembles the idea of copula, which models the marginal distribution and dependencies between random variables separately. Also, the proposed factor-analytic model provides a reduced decomposition for the high-dimensional correlation matrix. Finally, the learned correlation structure (for both, topics and words) is a byproduct of the model and does not introduce much computational complexity.

To Reviewer 2:

Q1: Section 2 is not well structured

A1: We agree and apologize for the confusion. We will reorganize section 2 (change the order of section 2.2 and 2.3), and improve its clarity in the revised version.

Q2: While the authors indicates that the proposed model is an alternative to dynamic topic models and correlated topic models, there is no comparison at all against these models in the experiment section

A2: The proposed model serves as an effective alternative to both the dynamic topic model and the correlated topic model, as it simultaneously captures the latent trajectories manifested, while also preserving the time-evolving correlation between topics and words. The interpretative quality of our results is very high, which is often of most interest to topic modeling. We may also effectively infer "jumps" in topic importance. However, as the marginal likelihood is nonparametric, a result is that we could not calculate the model evidence or other metrics based on it (e.g. held-out likelihood). This places a challenge for performing direct comparisons to other baseline models developed in the literature.

Q3: There is no evaluation of topic quality, like topic coherence, etc.

A3: A very good suggestion. We will examine this in future work. Thank you for pointing this out.

Regarding the reviewer’s other comments, we will double check the notations, improve the figures presentation, and add more discussions and interpretations about the results in the final version of the paper.

To Reviewer 3:

Regarding your concern on lacking objective evaluation, as mentioned, the nonparametric nature of our model, while providing significant advantages, is not easily directly compared to the likelihood/evidence in parametric models. Developing methods for performing such comparisons is something we intend to pursue in the future. We nevertheless feel that the interpretative quality of our model is already high and practically important.